# Non-canonical miRNA-RNA base-pairing impedes tumor suppressor activity of miR-16

Anaïs M Quéméner[1], Laura Bachelot[1,*], Marc Aubry[2,*], Stéphane Avner[3], Delphine Leclerc[2,4], Gilles Salbert[3], Florian Cabillic[5,6], Didier Decaudin[7,8], Bernard Mari[9], Frédéric Mouriaux[2,4], Marie-Dominique Galibert[1,10], David Gilot[1,2]

Uveal melanoma (UM), the most common primary intraocular tumor in adults, has been extensively characterized by omics technologies during the last 5 yr. Despite the discovery of gene signatures, the molecular actors driving cancer aggressiveness are not fully understood, and UM is still associated with very poor overall survival (OS) at the metastatic stage. By defining the miR-16 interactome, we revealed that miR-16 mainly interacts via non-canonical base-pairing to a subset of RNAs, promoting their expression levels. Consequently, the canonical miR-16 activity, involved in the RNA decay of oncogenes, such as *cyclin D3*, is impaired. This non-canonical base-pairing can explain both the derepression of miR-16 targets and the promotion of oncogene expression observed in patients with poor OS in two cohorts. miR-16 activity, assessment using our RNA signature, discriminates the patient's OS as effectively as current methods. To the best of our knowledge, this is the first time that a predictive signature has been composed of genes belonging to the same mechanism (miR-16) in UM. Altogether, our results strongly suggest that UM is a miR-16 disease.

## Introduction

Uveal melanoma (UM) is the most common primary intraocular tumor in adults. No effective treatment is currently able to counteract UM metastasis (Jager et al, 2020). UM carries mutually exclusive mutations that trigger overactivity of the Gαq pathway (G protein subunit alpha q [GNAQ], G protein subunit alpha 11 [GNA11], Phospholipase C Beta 4 [PLCB4], or Cysteinyl Leukotriene Receptor 2 [CYSLTR2]) (Robertson et al, 2018).

UM is considered to be a G protein-coupled receptor disease. However, additional genetic events occur, such as BES alterations, BRCA1-associated protein 1 (BAP1), Eukaryotic Translation Initiation Factor 1A X-Linked (EIF1AX) and Splicing Factor 3b Subunit 1 (SF3B1), and recurrent copy number variations (CNVs). UM is usually a diploid tumor with recurrent CNV of whole chromosomes or arms. Among these genome structural variations, monosomy 3 is the most frequent (~50% of cases). 1p loss, 1q gain, 6p gain, 6q loss, 8 gain, 8p loss, and 8q gain complete the UM genomic landscape. Monosomy 3 is clearly associated with a high risk of metastasis (Horsman et al, 1990; Bagger et al, 2014; Robertson et al, 2018; Shain et al, 2019). Although the loss of chromosome 3 induces the loss of *BAP1* (3p21.1), the role of the other genes located on chromosome 3 in tumor aggressiveness is not excluded. Gain of 8q is also common in UM (~50% of cases). It has been suggested that genes on chromosome 8q, such as *MYC*, and *POU Class 5 Homeobox 1* (*POU5F1*), and *Protein Tyrosine Phosphatase 4A3* (*PTP4A3*) may explain the poor overall survival (OS) of UM patients (Meir et al, 2007; Durante et al, 2020; Pandiani et al, 2021). *PTP4A3* has been shown to promote migration and invasiveness in an UM cell line, and high *PTP4A3* mRNA levels correlate with poor OSs of patients. It is important to note that the high level of *PTP4A3* mRNA (up-regulated in 66% of UM) is not merely a consequence of 8q gain (Laurent et al, 2011; Duciel et al, 2019). To date, the molecular mechanism promoting high expression levels of *PTP4A3* and *MYC* remains unsolved despite the deleterious consequences.

Although UM is considered to be a G protein–coupled receptor disease with BES alterations and CNVs, at least three other cellular processes seem to be frequently deregulated. These include the YAP pathway which is activated via Hippo-independent activation, splicing activity and translation initiation as a result of alterations in *BAP1*, Serine And Arginine Rich Splicing Factor 2 (*SRSF2*), RNA Binding Motif Protein 10 (*RBM10*) *EIF1AX*, and other genes

[1]University of Rennes, Centre National de la Recherche Scientifique (CNRS), Institut de Génétique et Développement de Rennes (IGDR) - UMR 6290, Rennes, France [2]INSERM U1242, University of Rennes, Rennes, France [3]SPARTE, University of Rennes, CNRS, IGDR - UMR 6290, Rennes, France [4]Service d'Ophtalmologie, CHU de Rennes, Rennes, France [5]NSERM U1241, Université Rennes, INRAE, Institut NuMeCan (Nutrition, Metabolisms and Cancer), Rennes, France [6]Laboratoire de Cytogénétique et Biologie Cellulaire, CHU Rennes, Rennes, France [7]Laboratory of Preclinical Investigation, Translational Research Department, Institut Curie, PSL Research University, Paris, France [8]Curie, Department of Medical Oncology, PSL Research University, Paris, France [9]Fédération Hospitalo Universitaire-OncoAge, CNRS, Institut de Pharmacologie Moléculaire et Cellulaire, Université Côte d'Azur, Valbonne, France [10]CHU Rennes, Service de Génétique Moléculaire et Génomique, Rennes, France

Correspondence: david.gilot@univ-rennes1.fr
*Laura Bachelot and Marc Aubry contributed equally to this work.

   

(Robertson et al, 2018; Shain et al, 2019). In 2018, an integrated analysis of 80 primary UMs was performed by The Cancer Genome Atlas (TCGA) to identify the deregulated pathways in this rare cancer with the end goal of finding druggable targets. Four mRNA signatures were generated based on tumor progression (Robertson et al, 2018). Other signatures have been generated (Harbour & Chen, 2013; Li et al, 2018; Luo et al, 2020) with few common genes. Recently, single-cell RNA sequencing results identified another RNA-signature called PC1. Using this signature, Bertolotto's team was then able to predict, with great accuracy, which patients would develop metastases (Pandiani et al, 2021; Strub et al, 2021). Partially overlapping signatures identified 5-Hydroxytryptamine Receptor 2B (*HTR2B*) as a marker of poor prognosis (Robertson et al, 2018; Weidmann et al, 2018; Le-Bel et al, 2019; Ni et al, 2019; Onken et al, 2021; Pandiani et al, 2021).

Gene expression patterns determine cell fate such as invasion capability; a critical process required for UM metastasis. miRNA shapes gene expression by inducing RNA decay via miRNA–RNA base-pairing. The factors determining miRNA binding to RNA are not fully understood. Biochemical characterization of miRNA targets (targetome) has indicated that miRNAs mainly induce RNA decay via the seed region of miRNA (nucleotides 2–7) and that in silico predictions are imperfect (Bartel, 2009).

Counterintuitively, increasing evidence indicates that miRNAs bind to a subset of RNA without inducing their decay. In this case, the base-pairing between the seed region of the miRNA and the RNA is imperfect. The biological role of these imperfect interactions is still a topic of debate. It has been suggested that these interactions are artefactual, rare, and not relevant because miRNA does not exert decay activity (Agarwal et al, 2015; Bartel, 2018). Nevertheless, experiments based on cross-linked immunoprecipitation (CLIP) and alternative methods, performed by different teams, have identified such interactions (Loeb et al, 2012; Helwak et al, 2013; Grosswendt et al, 2014; Luna et al, 2015). It has been hypothesized that the imperfect miRNA binding to RNA might hamper the miRNA decay activity (Bartel, 2009). In other words, miRNA can be inactivated when "sponged" or "sequestered" in RNA via imperfect binding (also known as non-canonical binding). This concept, initially described in *Arabidopsis thaliana*, relies on the competition between RNAs to bind to a limited amount of miRNA (Franco-Zorrilla et al, 2007). This model has been extended to eukaryotes as the competing endogenous RNA (ceRNA) model (Salmena et al, 2011). The interpretation of ceRNA results is still a topic of debate because the stoichiometry between miRNA and RNAs (sponges) is not (or rarely) investigated. Yet, the overexpression of synthetic constructs sponging miRNA (miRNA sponges) has been shown to effectively reduce the activity of miRNA in cancer and plant cells, strongly suggesting that miRNA sequestration is achievable, in particular experimental conditions. We summarized the criteria describing a miRNA sponge in 2017 (Migault et al, 2017). Briefly, a sponge should be highly expressed to sequester all the targeted-miRNA and ideally should contain several imperfect miRNA-binding sites per linear RNA molecule such as tyrosinase-related 1 (*TYRP1*) mRNA (Gilot et al, 2017). Convincingly, the sequestration of miRNA has also been illustrated in vivo on circular RNA (Kleaveland et al, 2018; Hanniford et al, 2020), confirming the physiological relevance of this type of miRNA regulation. To date, well-characterized miRNA

sponges have been rare in the literature because of the specific expression pattern of circRNA and the difficulty in predicting imperfect base-pairing between miRNA and sponges using current algorithms thought for linear RNA.

Because one copy of the *MIR16* gene is located on chromosome 3 and monosomy 3 is clearly associated with a high risk of metastasis, we evaluated the miR-16 expression level and activity of this tumor suppressor in UM. We hypothesized that the global miR-16 activity might be diminished in the case of monosomy and/or would be inactivated by sequestration on RNA via non-canonical binding.

## Results

### miR-16 is a potent tumor suppressor in UM

miR-16 is encoded by two intronic loci on the human genome. MIR16-1 is located on the intronic region of *Deleted in Lymphocytic Leukemia 2* (*DLEU2*) on chromosome 13 and MIR16-2 on the intronic region of *Structural Maintenance of Chromosomes 4* (*SMC4*) on chromosome 3 (Fig 1A). Both are transcribed into pri-miR-16-1 and pri-miR-16-2, then respectively processed into pre-miR-16-1 and pre-miR-16-2 to generate a similar product; miR-16 (Lagos-Quintana et al, 2001; Mourelatos et al, 2002). Because chromosome 3 monosomy is detected in more than 50% of patients with UM (Robertson et al, 2018), we postulated that the expression level of the tumor suppressor miR-16 might be reduced and consequently might impact tumor growth as observed for Chronic Lymphocytic Leukemia (LLC) patients (Calin et al, 2002) (Fig 1B). We confirmed a 50% decrease in miR-16 expression in samples from leukemia patients mainly because of the absence of pri-miR-16-1 synthesis (Fig 1C). Next, we examined the expression levels of mature miR-16 in patients with UM (TCGA cohort) according to the chromosome 3 copy number. Surprisingly, miR-16 expression was not altered by chromosome 3 monosomy in UM (Fig 1D). To strengthen these results, we used RT-qPCR to compare the miR-16 expression level in three UM cell lines with the miR-16 levels found in the 501Mel cell line (cutaneous melanoma). We had performed the latter in a previous study, obtaining an absolute quantification of miR-16 by Northern blot experiments (Gilot et al, 2017). These commercial cell lines harbour different mutations determined by Amirouchene-Angelozzi and colleagues and also by Jager and colleagues. MP41 is *GNA11* mutated, but the second event is still unknown. Mel202 and 92-1 cell lines are *GNAQ* mutated, whereas the second event is the mutation of *SF3B1* and *EIF1AX*, respectively (Amirouchene-Angelozzi et al, 2014; Jager et al, 2016). miR-16 is highly expressed in all three UM cell lines, and its expression seems independent of the chromosome 3 status in these selected cell lines. Normal human melanocytes have been used as controls (Fig 1E). Altogether, these results showed that the miR-16 expression level is not reduced in chromosome 3 monosomy UM tumors and cell lines.

However, miRNA expression does not always correlate with miR-16 activity. Sequestration of miR-16 by coding and non-coding RNA, referred to as miRNA-sponges, can dampen the miRNA activity as we demonstrated in cutaneous melanoma (Gilot et al, 2017). Repression of miR-16 target mRNA is thus alleviated, promoting in fine

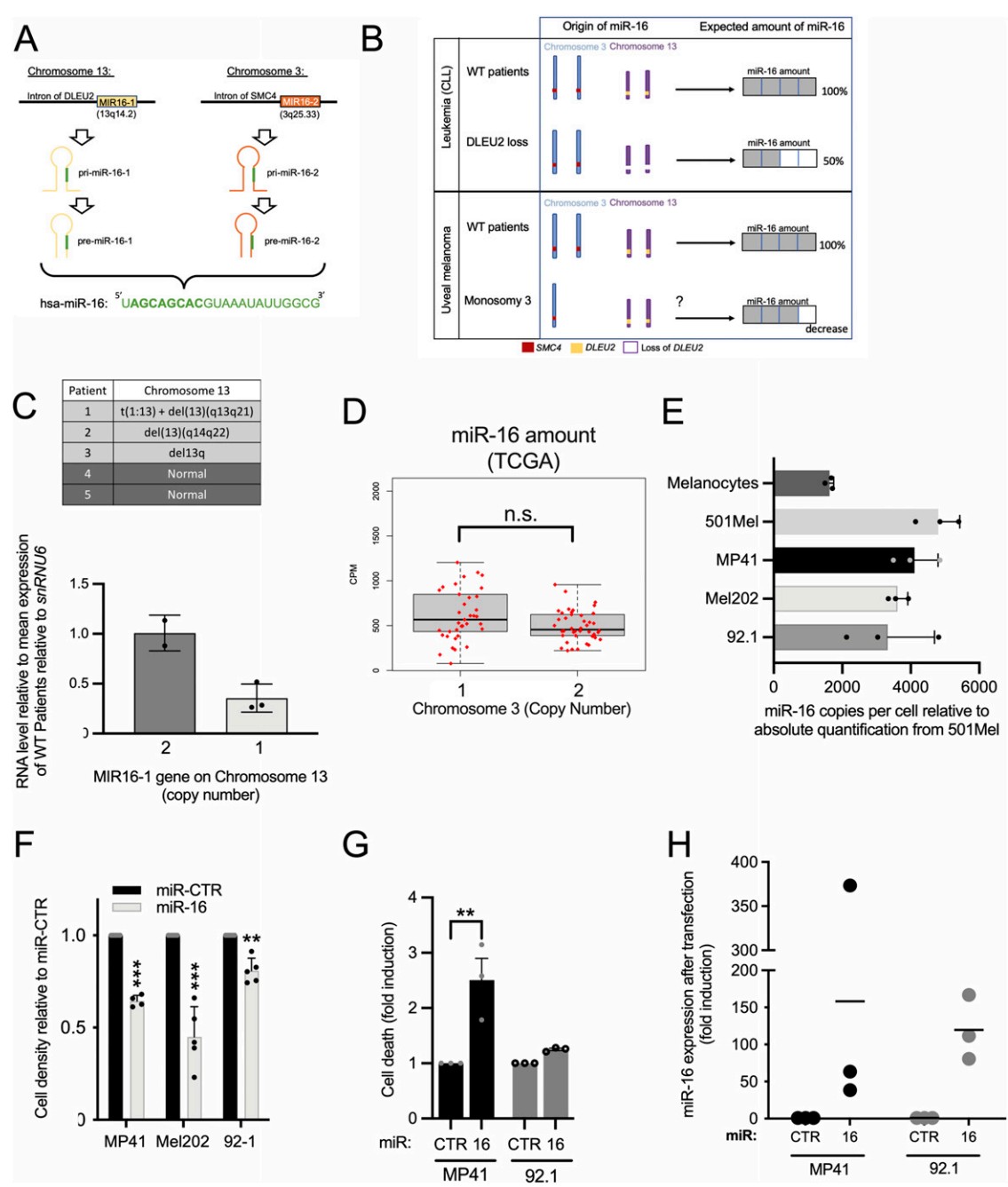

**Figure 1: miR-16 in uveal melanoma (UM).**
**(A)** Schematic representation of the genomic loci of miR-16 (MIR16-1 and MIR16-2) and miR-16 precursors: pri- and pre-miR-16 (1 and 2) and miR-16. The bolt region on the miR-16 sequence corresponds to the seed region of the miRNA. **(B)** Schematic representation of the expected amount of miR-16 according to the chromosomal status for the two miR-16 loci, for both leukemia and UM patients; (CLL, chronic lymphocytic leukemia). **(C)** Table of cytogenetic features of the chromosome 13 and miR-16 quantification for this five samples (CLL or healthy donors, n = 3 and n = 2, respectively). WT for healthy patients with two copies of chromosome 13. del for deletion. WT was set to a value of 1. Each histogram represents the mean + SD. **(D)** Boxplots of miR-16 expression according to the status of the chromosome 3 for UM patients from The Cancer Genome Atlas cohort (expressed in counts per million). Chromosome 3 monosomy n = 37 and disomy n = 42. Each histogram represents the mean + SD. n.s., nonsignificant. **(E)** Quantification of miR-16 expression in three UM cell lines (MP41, Mel202, and 92-1) and primary human melanocytes by RT-qPCR, compared with cutaneous melanoma cell line (501Mel). The absolute quantification (copy number) of miR-16 in 501Mel was determined by Northern blot in Gilot et al (2017). n = 3 biologically independent experiments for each cell line. Each histogram represents the mean + SD. **(F)** Cell density of MP41, Mel202, and 92-1 72 h after miR-16 transfection (transfection of synthetic miR-16 versus miR-CTR). n = 4, 5 and 5 biologically independent experiments, respectively. Each histogram represents the mean + SD; bilateral Student test (with non-equivalent variances) **P < 0.01; ***P < 0.001. **(G)** Fold induction of dead cells (apoptosis + necrosis; in % relative to miRNA control) in response to the miRNA transfection in MP41 and 92.1 cells (72 h). n = 3 biologically independent experiments. Each histogram represents the mean + SD; bilateral Student test (with non-equivalent variances) **P < 0.01. **(H)** miR-16 quantification by RT-qPCR after miRNA transfection in two UM cell lines. miR-16 values obtained in miRNA control transfected cells were set at 1 (endogenous miR-16 expression). n = 3 biologically independent experiments.

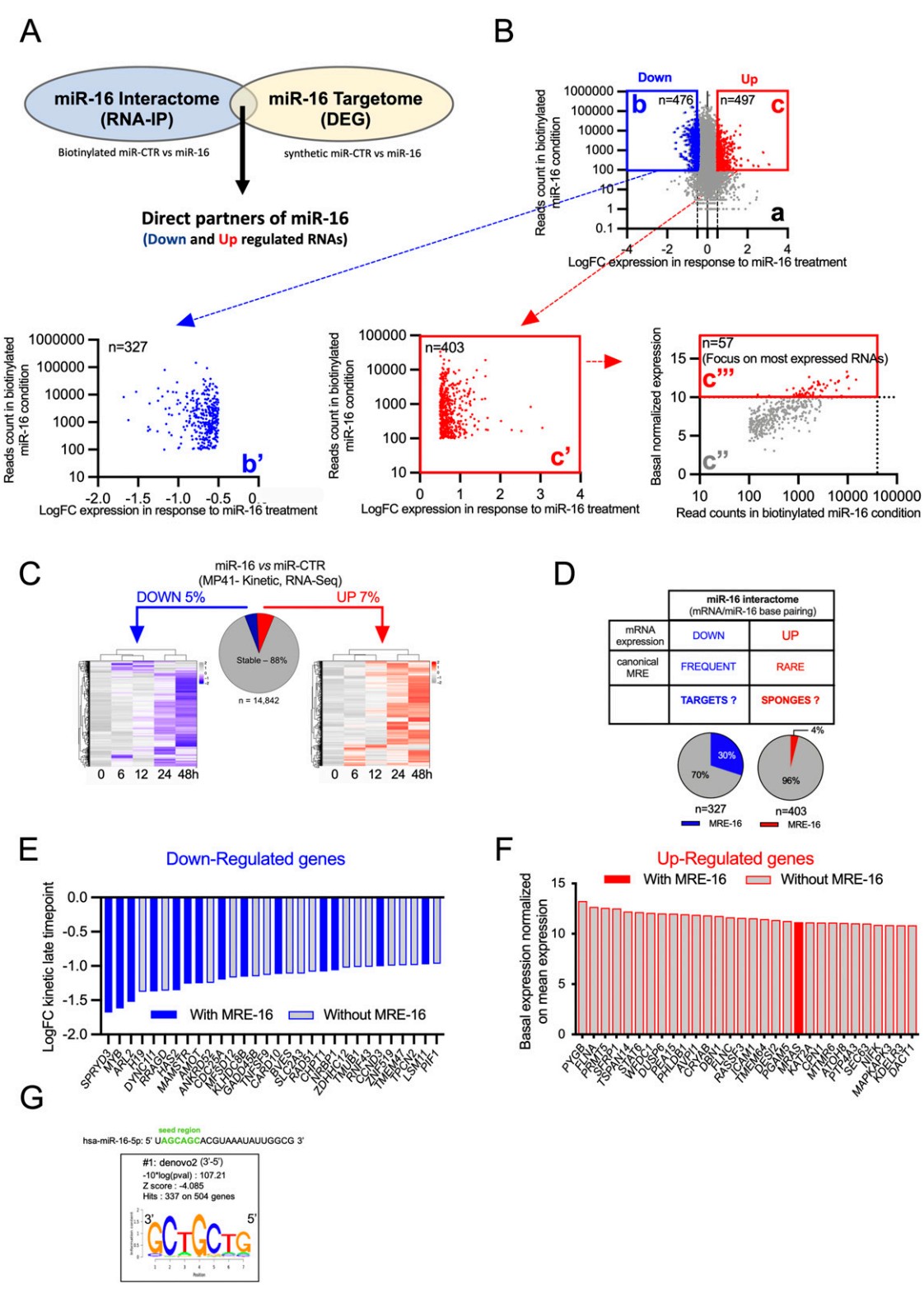

**Figure 2. miR-16 interactome.**
**(A)** Workflow to pinpoint direct partners of miR-16. To identify the miR-16–interacting mRNAs, a miRNA pull-down experiment followed by RNA sequencing was performed, and to identify the miR-16 targetome (up- and down-regulated genes), a gene expression analysis was performed by RNA sequencing after miR-16 transfection (kinetic analysis: 0, 6, 12, 24, and 48 h post-transfection). **(B)** Graph a: plot showing the miR-16 effects on miR-16–interacting RNAs (up or down-regulated). Representation of miR-16–interacting RNAs (reads count in biotinylated miR-16 condition [RNA-IP]) in function of their expression levels (fold change [FC]) after miR-16 transfection. Two clusters were delimited. The cluster b, in blue, corresponds to RNAs with a reads count > 100 and down-regulated at RNA level with a fold change < −0.5

tumor growth (Karreth & Pandolfi, 2013; Migault et al, 2017). We hypothesized that a comparable mechanism might mediate miR-16 inactivation in UM. To test this hypothesis, we first investigated the tumor suppressor activity of miR-16 in UM cells by elevating miR-16 expression levels. UM cell density decreased specifically after 72 h following transfection of synthetic miR-16 (Fig 1F–H), suggesting that miR-16 indeed acts as a tumor suppressor in human UM.

### Definition of miR-16 interactome

To identify the RNAs involved in miR-16 sequestration, and consequently the dysregulated target RNAs, we defined the miR-16 interactome (mRNA interacting with biotinylated miR-16) using RNA pull-down. We confronted this result with transcriptomic profiling in response to synthetic miR-16 transfection in UM cells that define the miR-16 targetome (MP41) (Fig 2A–D and Table S1A). To discard artefactual interactants due to nonspecific background inherent to all biotinylated experiments (Lal et al, 2011; Tan & Lieberman, 2016; Dash et al, 2018), we analyzed our data in two steps. First, we plotted the log fold change (logFC) expression after miR-16 transfection and the detection of these RNAs in pull-down using biotinylated miR-16 (Fig 2B). miR-16 interactants were arbitrarily defined (logFC < −0.5 or >0.5 and a number of reads superior to 100 in the pull-down assay). Thus, we defined two windows (b and c) containing, respectively, 476 and 497 potential miR-16 interactants. Second, we removed all highly detected RNAs (>500 reads) in the biotinylated miR-CTR condition, considering them as false-positive candidates. Finally, we obtained two groups of miR-16 partners. First, 327 miR-16 interactants for which expression levels decreased in response to miR-16 transfection, strongly suggesting that these RNAs (b' window) correspond to miR-16 targets (targetome) in our model. Some of these miR-16 targets have already been identified as a miR-16 target in the literature, including *Cyclin D1* (*CCND1*), *Cyclin D3* (*CCND3*), and *WEE1*, validating our experimental workflow (Liu et al, 2008; Cai et al, 2012; Lezina et al, 2013). Second, we applied the same filter for the up-regulated RNAs and we obtained an additional cleaned list of 403 miR-16 interactants (window c' in Fig 2B) (list available in Table S1A).

The biological function of up-regulated miRNA interactants remains unclear in the literature even though they have already been highlighted by different teams (Vasudevan et al, 2007). Here, we postulated that down-regulated miR-16 interactants are "real miR-16 targets" and up-regulated miR-16 interactants might correspond to "potential miR-16 sponges" (Fig 2D). Because a miRNA-sponge (mRNA) should be highly expressed and contain non-canonical miRNA-binding site(s) avoiding its decay (Gilot et al, 2017), we focused on highly up-regulated miR-16 interactants (basal normalized expression level > 10) (window c'' in Fig 2B), identifying 57 potential miR-16 sponges in UM (list available in Table S1A). As expected, 30% of down-regulated RNAs (miR-16 targets) contained predicted canonical miR-16–binding sites (microRNA response element [MRE]-16) (Fig 2D, E, and G) (Agarwal et al, 2015) and only 2% of the up-regulated miR-16 interactants did (Fig 2D and F), suggesting that miR-16 base-pairing to potential sponge RNAs might be non-canonical.

To explain why only 30% of miR-16 targets display MRE-16 predicted by TargetScan 7.2, we further studied the miR-16–binding sites on down-regulated RNAs by examining the complete sequence of these RNAs and not only their 3'UTR using Cistrome SeqPos motif analysis. Interestingly, the most meaningful motif resembles the miR-16–binding site (motif #1: 3'-<u>GCTGCTG</u>-5' [underlined sequence is complementary to miR-16 sequence: 5'_u<u>AGCAGC</u>ac——_3']). Of note, this motif is not retrieved exactly in up-regulated miR-16 interactants (potential sponge RNA) (Fig S1). However, motif #1, identified on sponge RNAs; -<u>GCTG</u> (or T or A) <u>CT</u>-, is quite similar to motif #1, identified on miR-16 targets, except for the nucleotide in position 5 (G, T, or A). This result suggests that a bulge may be created when the seed sequence of miR-16 binds to the sponge RNAs. This type of bulge is usually described to strongly reduce or abolish the RNA decay mediated by a miRNA (Agarwal et al, 2015; Kim et al, 2016a), in accordance with our hypothesis.

### miR-16 modulates cell fate by targeting several RNAs in UM

Before exploring the role of these miR-16 interactants, we verified that they were not cell-line restricted. Thus, using the same approach (synthetic miR-16 transfection), we validated 19 candidates in two other UM cell lines (Mel202 and 92-1) (Fig 3A and B). We observed a down-regulation of these miR-16 targets in all cell lines except *MYB* in the Mel202 cell line. We obtained comparable results

---

(n = 476 genes). The cluster c, in red, corresponds to RNAs with a reads count (RNA-IP) > 100 and up regulated with a fold change > 0.5 (n = 497 genes). Graph b': represents same genes of the graph b without those suspected to be false positive because of their detection with biotinylated miR-CTR (threshold 500 reads in the RNAseq, for miR-CTR) (n = 327 genes). Graph c' represents same genes of the graph c without those suspected to be false positive because of their detection with biotinylated miR-CTR (threshold 500 reads in the RNAseq for miR-CTR) (n = 403 genes). Graph c'' in grey: the same selected genes of the graph c'' but they are represented according to their basal expression by pull-down enrichment (miR-16 − miR-CTR). The c''' cluster represents only genes with a basal expression > 10 (normalized expression). This workflow identified 57 potential sponges. **(C)** Heat map representing the differential transcriptomic response after transfection of miR-16 versus miR-CTR in MP41 cell line (0 h = starting time point, 6, 12, 24, and 48 h post-transfection). MP41 transcriptome (n = 14,842 genes) are divided in three populations. By comparing miR-16 condition versus miR-CTR at early time point (6–12 h) and late time (24–48 h), three populations have been identified: stable genes ~88%, down-regulated (LogFC < −0.5) genes ~5% and up-regulated (LogFC > 0,5) genes ~7%. Left heat map illustrating the down-regulated RNAs in response to miR-16 transfection in MP41 (0, 6, 12, 24, 48 h). Right heat map illustrating the up-regulated RNAs (Table S1). **(D)** Table describing the expected miR-16 interactome (mRNAs interacting with miR-16) in function of the experimental workflow detailed in Fig 2A. miR-16–interacting mRNAs have been considered as potential targets or potential sponges (respectively, down- and up-regulated genes after miR-16 transfection). MRE for miRNA response element. Pie charts indicate the percentage of mRNAs ("targets or sponges") harbouring at least one MRE-16 (predicted by TargetScan 7.2) (Agarwal et al, 2015). **(E)** Down-regulated regulated genes (potential miR-16 targets) are ordered according to the level of the down-regulation at the late time point. Only the top 30 targets are represented. Solid blue bars indicate at least one canonical MRE-16 predicted by TargetScan 7.2. **(F)** Basal expression of the 30 most expressed genes at the basal level (from the selected genes represents in the graph c''' [Fig S3B]). Solid red bars indicate at least one canonical MRE-16 predicted by TargetScan 7.2. **(G)** Consensus sequence of miRNA-binding site motif enriched in cluster of down-regulated RNAs after miRNA pull-down in MP41 (analyzed by Cistrome SeqPos [Liu et al, 2011]).

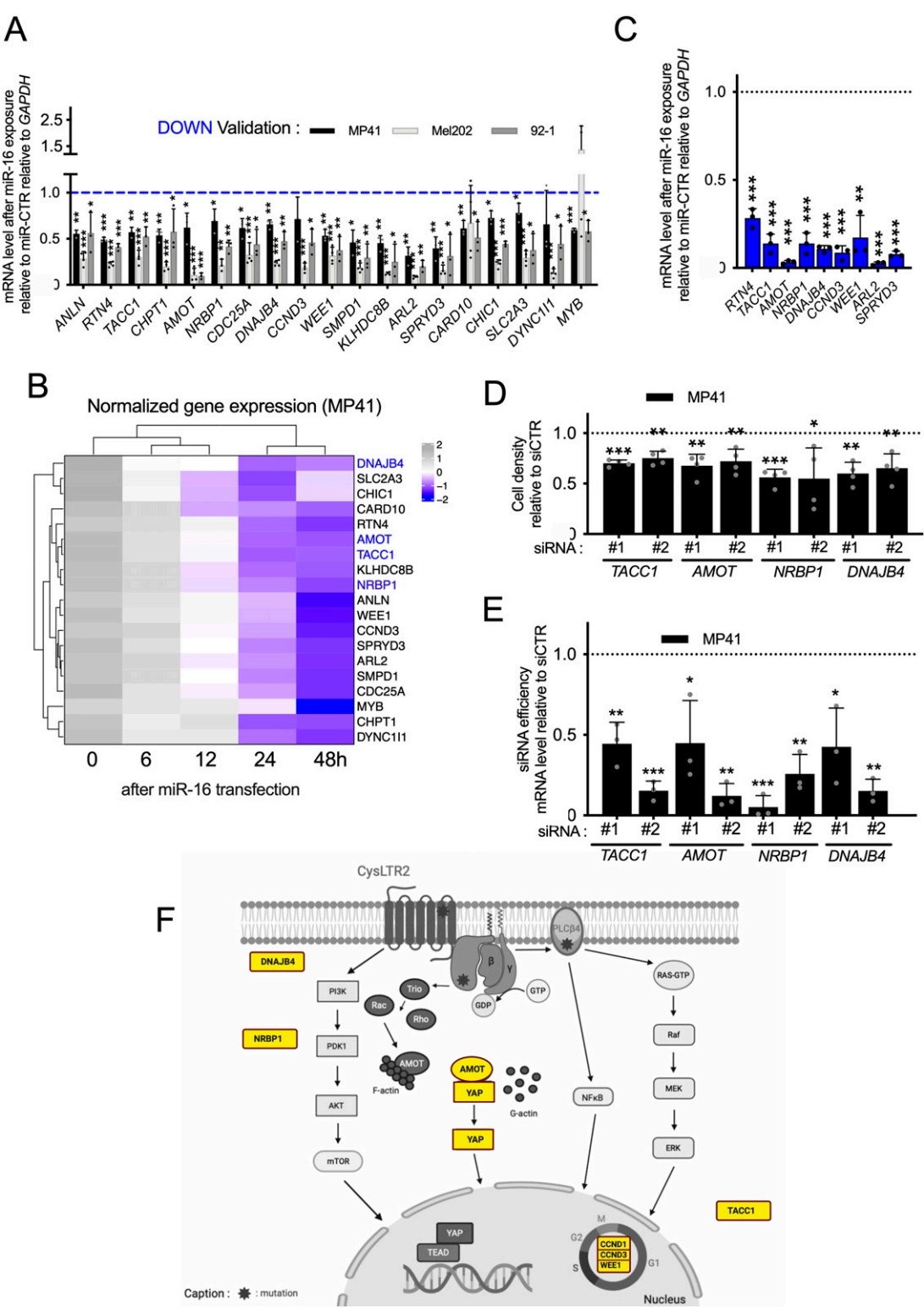

**Figure 3. miR-16 modulates cell fate by targeting several RNAs in uveal melanoma.**

**(A)** mRNA expression levels of potential miR-16 targets, 72 h after transfection of miR-16 relative to miR-CTR in MP41, Mel202, and 92-1 cells. n = 3, 4, and 3 biologically independent experiments, respectively. Each histogram represents the mean ± SD; bilateral Student test (with non-equivalent variances) *P < 0.05; **P < 0.01; ***P < 0.001. **(B)** Part of Fig 2C; heat map illustrating the selected genes of Fig 3A, analyzed by RNAseq, after miR-16 transfection in MP41 cells (kinetic analysis: 0, 6, 12, 24, and 48 post-transfection). **(C)** mRNA expression levels of potential miR-16 targets (down-regulated genes), 72 h after transfection of miR-16 or miR-CTR in HCT116 DROSHA KO. n = 3 biologically independent experiments. Each histogram represents the mean ± SD; bilateral Student test (with non-equivalent variances) **P < 0.01; ***P < 0.001. **(D)** Cell density of MP41 cells after *TACC1*, *AMOT*, *NRBP1*, and *DNAJB4* depletion by two different siRNAs (#1 and #2) relative to siCTR (n = 4 biologically independent experiments) 72 h

in the HCT116 colon cell line in which almost all miRNAs are lost because of DROSHA KO (Luna et al, 2015), including endogenous miR-16 (Fig 3C). Altogether, these results suggest that the miR-16 targets identified in the MP41 cell line are highly conserved in the same cancer models (UM), despite the diversity of driver mutations and chromosome abnormal copy number that characterize these three cell lines. Moreover, the miR-16 targetome is, at least in part, common to other models such as those in the colon.

Next, we investigated how miR-16 reduces melanoma cell survival/proliferation. Even though we confirmed that miR-16 targets cell cycle regulators such as *CCND3* and *WEE1* (Fig 3A), we investigated the biological consequences of the RNA decay of less-characterized RNA in UM. We thus performed depletions of four miR-16 targets (Angiomotin [*AMOT*], *Transforming Acidic Coiled-Coil Containing Protein 1* [*TACC1*]; *Nuclear receptor-binding protein* [*NRBP1*]; *Dna J Heat Shock Protein Family Hsp40 Member B4*, and [*DNAJB4*]). We selected these candidates according to decreasing expression during the kinetic experiment (Fig 3B), MRE-16 identified in their 3′UTR, and the lack of a clear connection between these RNAs and UM. We showed that depletion of *AMOT*, *TACC1*, *NRBP1*, or *DNAJB4* reduced cell density in UM cells (Fig 3D and E).

Altogether, our results suggest that miR-16 reduces melanoma cell proliferation by targeting different essential pathways/processes, including cell cycle regulators (Fig 3F). It is tempting to postulate that the loss of miR-16 activity due to miR-16 sequestration could promote cell proliferation by derepressing these miR-16 targets.

## Non-canonical miR-16 binding to miR-16 interactants promotes both RNA translation and miR-16 sequestration

Because the function of up-regulated miR-16 interactants is poorly studied, we first validated their miR-16–mediated increase in other cell lines (Mel202, 92-1 and HCT116 KO DROSHA) as done for miR-16 targets. For most of the tested RNA, we validated their increase after miR-16 transfection in at least one other cell line. Among them, some seem to be cell line specific (Fig 4A and B). In addition, 50% of the potential sponge mRNAs tested were still increased after miR-16 transfection in DROSHA knock-out cells in which almost all miRNAs, including miR-16, are lost, suggesting that miR-16 acts directly on these sponges rather than through competition with another miRNA involved in sponge decay.

Next, we assessed the translational consequence of miR-16 binding on these up-regulated miR-16 interactants because miRNA binding on canonical binding sites commonly provokes a translation blockade and induces RNA decay via seed base-pairing (Bartel, 2004). Here, we focused our attention on glycogen phosphorylase B (*PYGB*), defined as our best miR-16 sponge candidate because of its high expression level in UM (Fig 2F) and its increased response to miR-16 transfection in three UM models (Fig 4A). We showed that both *PYGB* mRNA and protein increased after miR-16 transfection in three UM cell lines (Fig 4A and C).

To elucidate the molecular mechanism, explaining the translation up-regulation mediated by miR-16, we looked for miR-16–binding sites on *PYGB* mRNA using the RNAhybrid webtool (Krüger & Rehmsmeier, 2006). It uses the energy needed to predict interaction between two RNAs. Two potential non-canonical miR-16–binding sites with high energy were found on *PYGB* mRNA (Fig 4D). To determine whether these two non-canonical miR-16–binding sites are involved in the PYGB protein up-regulation by miR-16, we cloned these sequences (sites 1 or 2, wild-type or mutated [WT or MUT]) fused with the luciferase coding sequence. The translation effectiveness of these chimeric RNAs was estimated by assessing luciferase activity. As a control, we used the luciferase coding sequence fused to canonical MRE-16 (from *CCND1*) (Fig 4E). As expected (Bonci et al, 2008), we found decrease in luciferase activity after miR-16 transfection for this canonical seed base-pairing. In contrast, the non-canonical miR-16–binding sites promoted higher luciferase activity compared with mutated sequences (Fig 4E and F).

Altogether, our results strongly suggest that miRNA/RNA base-pairing determines RNA stability and the translation rate; and a non-canonical MRE can promote translation, in contrast to a canonical MRE.

To further challenge the miR-16 sequestration hypothesis while preserving the stoichiometry between miR-16 and its interactome, we next depleted *PYGB* mRNA and quantified endogenous miR-16 targets (Fig 4G and H) selected as a function of: (*i*) a miR-16–dependent mRNA decay (Fig 3), (*ii*) presence of a predicted MRE-16 in their 3′UTR (Fig 2), and (*iii*) a decrease in MP41 cell density after their depletion (Fig 3D). Because the depletion of only one miR-16 sponge (*PYGB*) was followed by a moderate decrease in several miR-16 target RNAs (Fig 4G and H), it is tempting to conclude that miR-16 sequestration involves several RNAs with non-canonical MRE-16. This model of sequestration may explain why we identified 57 potential miR-16 sponges in UM (Fig 2B).

## miR-16 interactome predicts UM metastasis and survival

Our model is based on a constant level of miR-16. We investigated whether a high level of miR-16 sponges might be associated with a loss of canonical miR-16 activity and consequently associated with a poor OS of patients. We demonstrated that quantification of 57 miR-16 sponge candidates effectively predicted survival in UM patients (TCGA cohort), reflecting metastasis risk (Fig 5A). Unsupervised gene expression analysis identified two clusters: light and dark grey (cluster 2 and 1, respectively) (Fig 5A). These clusters are highly correlated with those defined by TCGA. Remarkably, the miR-16 expression level was comparable in the two groups, supporting the sponging hypothesis (Fig 5B). In accordance with our hypothesis, we showed that a high level of miR-16 sponges is associated with a dismal survival (Fig 5C) and that miR-16 targets are derepressed in cluster 1 (Figs 5C and S2). Altogether, these results indicate that miR-16 activity (assessed using miR-16 sponges and target expression levels) is a useful marker for

after transfection. Each histogram represents the mean ± SD; bilateral Student test (with non-equivalent variances) *$P < 0.05$; **$P < 0.01$; ***$P < 0.001$. **(E)** Efficiency of siRNA used for Fig 3D (evaluated by RT-qPCR, 72 h post-transfection). Two different siRNAs (#1 and #2)/gene. n = 3 biologically independent experiments. Expression relative to siCTR, *$P < 0.05$; **$P < 0.01$; ***$P < 0.001$. **(F)** Scheme summarized the most frequent genetic alterations found in uveal melanoma and the potential roles of several miR-16 targets in these deregulated pathways (Created with BioRender.com).

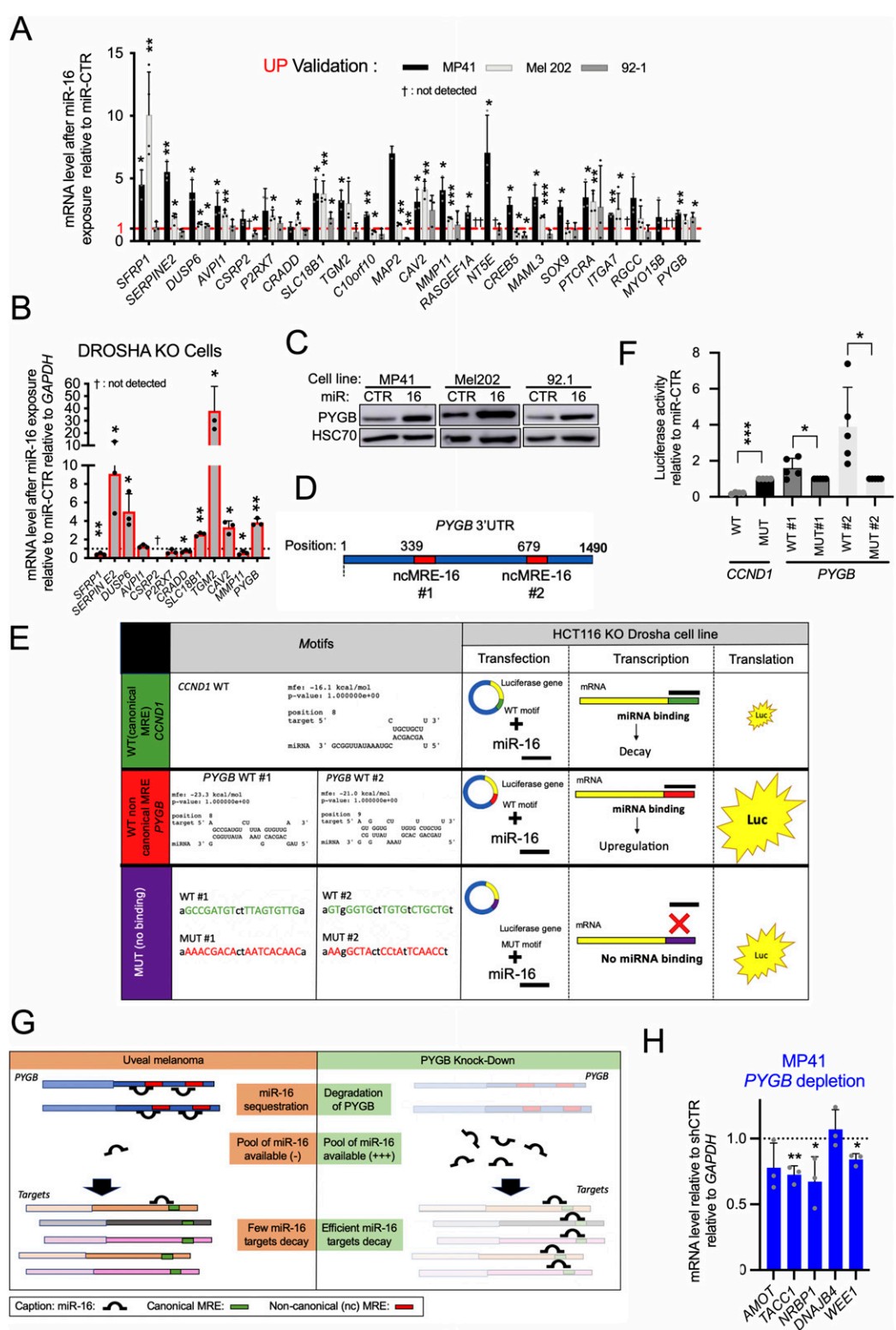

**Figure 4. miR-16 is sequestered on non-canonical miR-16–binding sites.**
**(A)** mRNA expression levels of potential miR-16 sponges (up-regulated genes), 72 h after transfection of synthetic miR-16 relative to miR-CTR in MP41, Mel202, and 92-1 cells. n = 3, 4, and 3 biologically independent experiments, respectively. Each histogram represents the mean ± SD; bilateral Student test (with non-equivalent variances) *P < 0.05; **P < 0.01; ***P < 0.001; †: Ct > 40. **(B)** mRNA expression levels of potential miR-16 sponges 72 h after transfection of synthetic miR-16 relative to miR-CTR in HCT116 DROSHA knock-out cells. n = 3 biologically independent experiments. Each histogram represents the mean ± SD; bilateral Student test (with non-equivalent variances) *P < 0.05; **P < 0.01; †: Ct > 40. **(C)** Protein expression levels of PYGB in MP41, Mel202, and 92-1 cell lines after miRNA transfection (72 h, miR-CTR versus miR-16). The picture is

clinicians, in contrast to miR-16 expression. Because 57 RNAs are too numerous to be exploited clinically, we developed a risk model (Luo et al, 2020) (Fig 6A and Table S2), identifying four RNAs useful for predicting the OS of patients with UM (signature S4: Figs 6B and S3). The S4 signature's ability to predict survival was confirmed in an independent cohort (n = 63; GSE22138) (Laurent et al, 2011) (Fig 6C).

Taken together, our results demonstrated that abnormal (non-canonical) miR-16 activity is associated with a poor clinical outcome in UM.

# Discussion

Gene expression is fine-tuned by miRNA at the post-transcriptional level. Alteration of miRNA expression or activity is associated with numerous diseases. We also now know that miRNA sequestration is an important event. Thus, it seems crucial to quantify a miRNA activity as much as its expression in the assessment of the role of miRNA in cancer.

Here, we characterize a molecular mechanism explaining the loss of tumor suppressor activity of miR-16 (loss of brake effect), which is associated with metastasis risk and poor OS in UM (Figs 5 and 6). Instead of promoting RNA decay of miR-16 targets such as cell cycle regulators (*CCND3*, *CCND1*, and *WEE1*), non-canonical miR-16 activity mediates the expression of pro-tumoral genes (acceleration effect). These include *PTP4A3* and *HTR2B* (Laurent et al, 2011; Le-Bel et al, 2019; Onken et al, 2021).

Canonical activity (mRNA decay) of the tumor suppressor miR-16 is impaired by miR-16 sequestration on non-canonical MRE-16. We identified, using complementary experiments, potential miR-16 sponge RNAs (defined as up-regulated miR-16 interactants). Importantly, we showed that non-canonical miR-16 binding on these mRNAs promotes their accumulation. Additional experiments are needed to explain this phenomenon. It is tempting to postulate that miR-16 on non-canonical MRE-16 might recruit FMR1 Autosomal Homolog 1 (FXR1), promoting RNA circularization and non-canonical translation (Vasudevan et al, 2007; Bukhari et al, 2016). This mechanism might explain the mRNA up-regulation in response to miR-16 binding and the increase in protein observed in *PYGB*. In addition, this circularization might thus increase the sponge activity of linear RNA because the most potent miRNA sponges are probably circRNAs (Guo et al, 2014). In accordance with this

hypothesis, a recent study demonstrated that miRNAs loaded in Ago2 are enriched in the 3'UTR of several RNAs without inducing their decay. Authors have clearly demonstrated that *MYC* RNA levels are increased in response to miRNA binding (Chu et al, 2020). Altogether, our results suggest that non-canonical miR-16 activity led to gain-of-function of pro-tumoral genes such as *PTP4A3* and *HTR2B*. These mRNAs are overexpressed in most UM cases associated with a poor clinical outcome, and could be targeted using antisense oligonucleotides. Thus, we suggest that miR-16 can exert pro- or anti-tumoral activity depending of its base-pairing to RNA.

The current concept of competition between RNAs to bind miRNA is still a topic of debate (Smillie et al, 2018). Here, we showed that miR-16 sequestration is probably carried out by several RNAs in UM because single *PYGB* knock-down induces a modest decay of miR-16 targets. This is in accordance with the identification of 57 potential miR-16 sponges in UM. However, additional biochemical experiments are need to further characterize the miR-16 interactions with these potential miRNA sponges as well as the potential miRNA targets identified here.

The clinical relevance of our results has been demonstrated using different UM cohorts. Our sponge signature (57 genes) and the signature S4 are therefore able to predict, with great accuracy, which patients will develop metastasis, as well as the PC1 signature established from single-cell RNA sequencing results (Pandiani et al, 2021; Strub et al, 2021). It is of interest that few genes are common to both of these two effective signatures. For our signature, all 57 genes are potential miR-16 interactants. To the best of our knowledge, this is the first time that a predictive signature has been composed of genes belonging to the same mechanism (miR-16) in UM.

In conclusion, our results highlight the need to update the current models explaining miRNA activity and its role in gene regulation and diseases.

# Materials and Methods

### Cell culture

MP41 cell line was obtained from Decaudin's laboratory at Curie Institute, Paris, France. Mel202 and 92.1 cell lines were obtained from European Collection of Authenticated Cell Cultures (ECACC)

representative of n = 3, 2, and 3 biologically independent experiments, respectively. **(D)** Schematic representation of the 3'UTR of *PYGB* mRNA containing two non-canonical MRE-16 predicted by RNAHybrid. **(E)** Biological function of non-canonical MRE-16. On the left side: predicted base-pairing between *PYGB* mRNA and miR-16 using RNAhybrid (Rehmsmeier et al, 2004). Base-pairing was evaluated for wild-type (WT) MRE-16 (non-canonical MRE-16 #1 and #2) from *PYGB* mRNA. On the right side: schematic representation of the luciferase assay. Canonical MRE-16 (from CCND1, [Liu et al, 2008]) has been used as positive control (miR-16 induced the decay of a mRNA harbouring a canonical MRE-16). The two non-canonical miR-16–binding sites of *PYGB* have been cloned in fusion with the luciferase coding sequence. The translation efficiency of these chimeric RNAs is estimated by assessing the luciferase activity. **(F)** Luciferase assay assessing the effect of synthetic miR-16 on these chimeric RNAs in HCT116 KO DROSHA cell line. Canonical MRE-16 (from CCND1, [Liu et al, 2008]) has been used as positive control (as attended miR-16 induced the decay of a mRNA harbouring a canonical MRE-16). MUT, mutated; WT, wild type (n = 5 biologically independent experiments). Each histogram represents the mean ± SD; bilateral Student test (with non-equivalent variances) *P < 0.05, ***P < 0.001. **(G)** Hypothetical scheme explaining the miR-16 displacement is response to miR-16 sponge depletion. Sequestered miR-16 on *PYGB* mRNA are released from *PYGB* and reached other miR-16–binding sites (on other RNAs including miR-16 targets). Based on our hypothesis, the expression levels of these targets should thus decrease. *PYGB* mRNA has been selected because it is the most expressed sponge identified in this study. Here, the stoichiometry between miR-16 and miR-16–interacting RNAs is preserved (no miR-16 transfection). **(H)** mRNA expression levels of miR-16 targets: *AMOT*, *TACC1*, *NRBP1*, *DNAJB4*, and *WEE1* after *PYGB* mRNA depletion in MP41 cells (shPYGB relative to shCTR) (n = 3 biologically independent experiments). Each histogram represents the mean ± SD; bilateral Student test (with non-equivalent variances); *P < 0.05; **P < 0.01.
Source data are available for this figure.

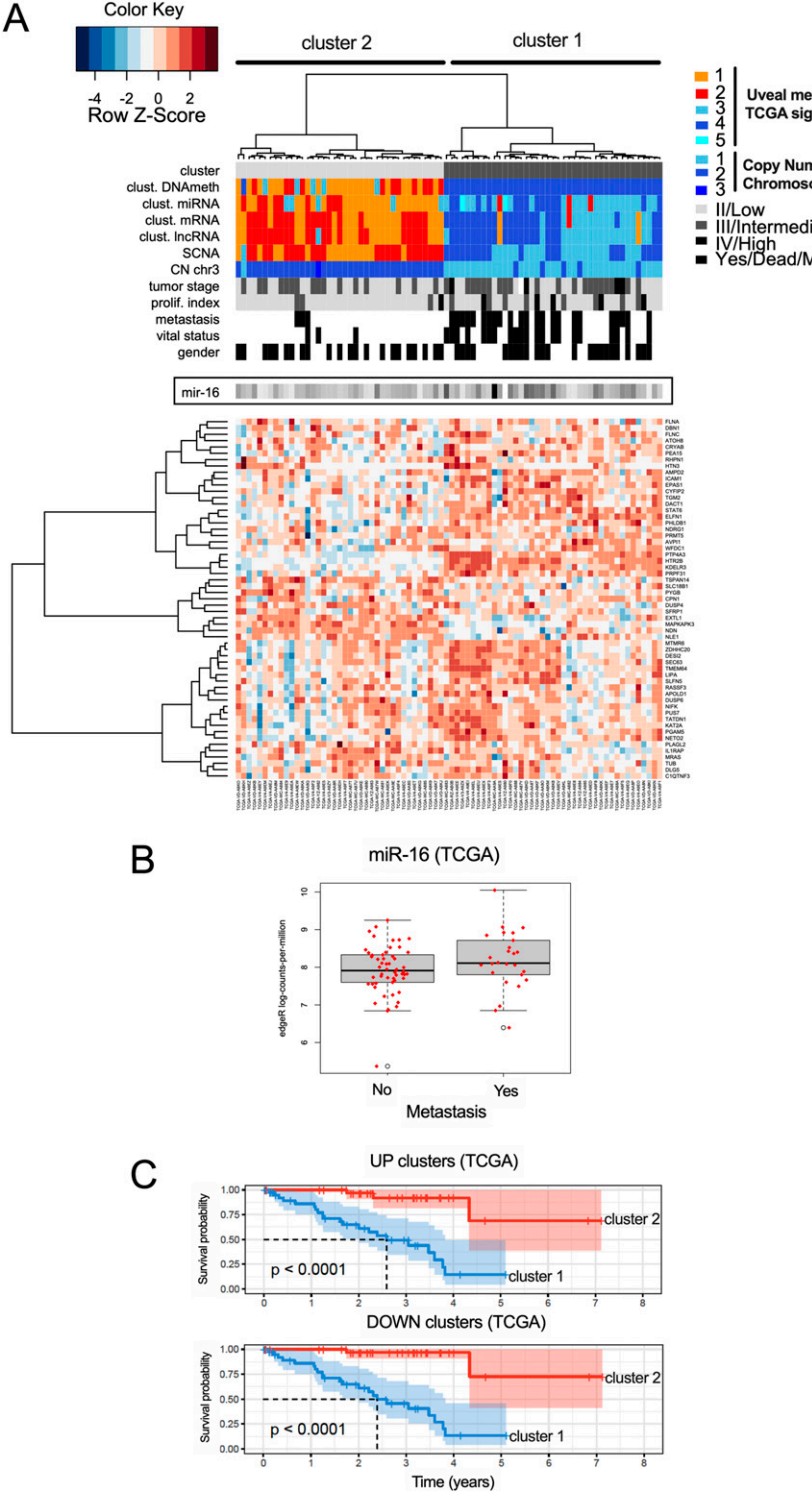

**Figure 5. miR-16 availability defines two signatures predicting the prognostic of uveal melanoma patients.**
**(A)** Heat map depicting the expression levels of potential 57 sponge RNAs—The Cancer Genome Atlas (TCGA) cohort of uveal melanoma. Unsupervised gene expression analyses identified 2 clusters: light and dark grey (clusters 2 and 1, respectively). Cluster 1 is associated with poor clinical outcome (chromosome 3 monosomy, metastasis). Moreover, cluster o1 overlaps with the TCGA signatures (miRNA, mRNA, lncRNA and DNA methylation) previously associated with poor clinical outcome. CN for copy number. **(B)** Boxplot comparing the amount of miR-16 in patients who developed metastases or not (TCGA cohort). No significant difference was found. **(C)** Determination of overall survival curves by the Kaplan–Meier analysis based on clusters 1 and 2. The difference in survival between groups is reported (log-rank test *P*-value). KM analyses have been performed for potential miR-16 sponges RNA and targets RNA according to the clusters 1 and 2 defined in Figs 5A and S2A and Table S1.

(Merck). 501Mel cell line was obtained from American Type Culture Collection (ATCC). HCT116 WT and HCT116 KO DROSHA (Kim et al, 2016b) cell lines were obtained from Korean Collection for Type Cultures (KCTC), Microbial Resource Center. All cell lines were maintained in humidified air (37°C, 5% CO$_2$). UM cell lines were maintained in RPMI-1640 medium (Gibco, Thermo Fisher Scientific) supplemented with 20% FBS (EurobioScientific) and 1% penicillin–streptomycin antibiotics (Gibco, Thermo Fisher Scientific). 501Mel was maintained in RPMI-1640 medium (Gibco, Thermo Fisher Scientific) supplemented with 10% FBS (EurobioScientific) and 1% penicillin–

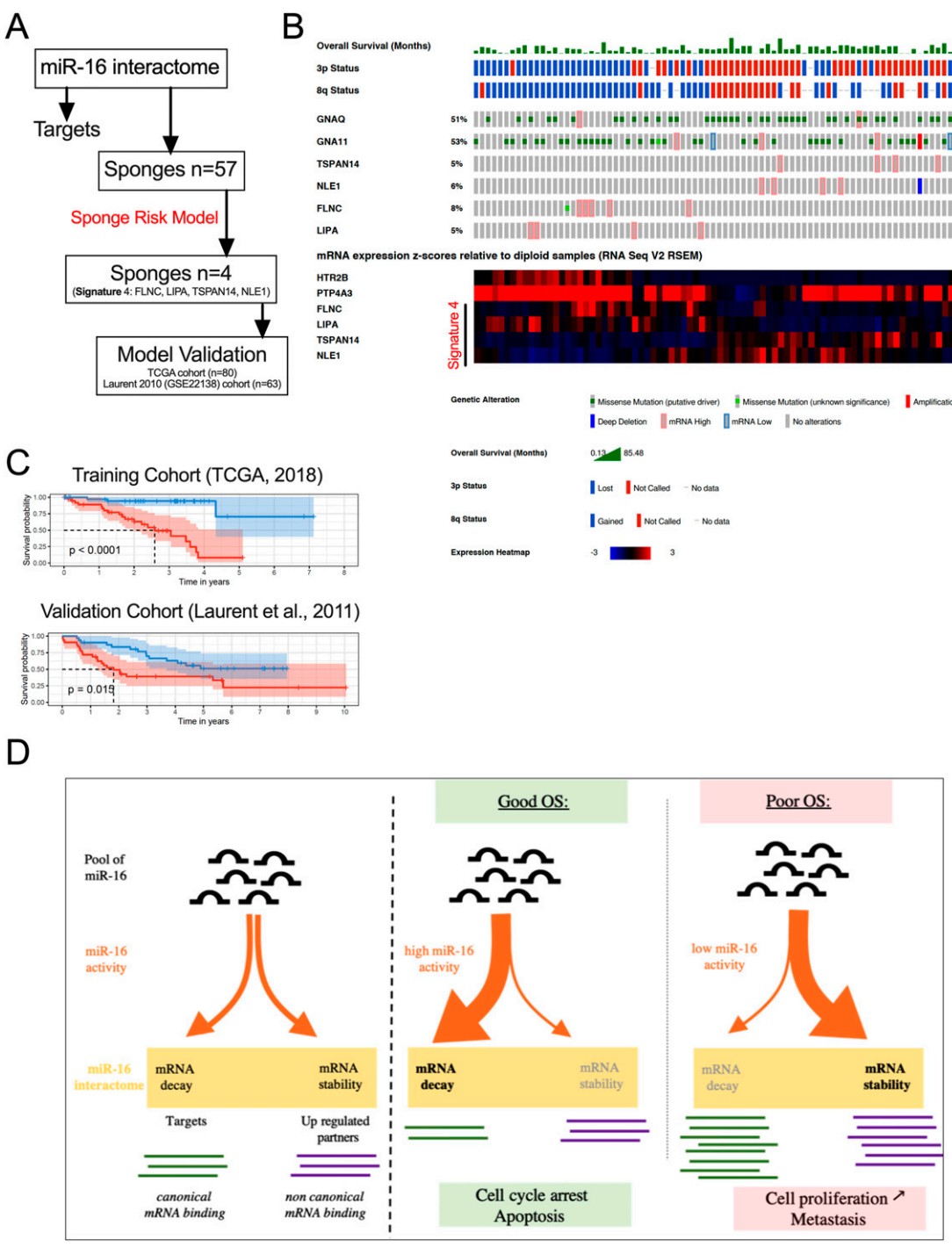

**Figure 6.  Signature S4 predicts overall survival (OS) for uveal melanoma (UM).**
**(A)** The sponges risk model workflow identifying four sponge RNAs (the signature S4). The Cancer Genome Atlas (TCGA)-UVM cohort has been used as a training cohort and the GEO dataset GSE22138 as a validation cohort. We trained an optimal multi-gene survival model based on the expression of the sponges in the training cohort by selecting survival-associated genes with the *rbsurv* R package using 1,000 iterations. Briefly, this package allows a sequential selection of genes based on the Cox proportional hazard model and on maximization of log-likelihood (see the Materials and Methods section and Table S2). Risk scores were determined using classical Cox model risk formulae with a linear combination of the gene expression values weighted by the estimated regression coefficients. The risk cutoff was set to the median of the linear predictor. The Kaplan–Meier method was used to estimate the survival distributions. Log-rank tests were used to test the difference between survival groups. Analyses were carried out with the *survival* and *survivalROC* R packages. **(B)** Genetic alterations described in the TCGA cohort of UM from Cbioportal for our four potential sponges (*TSPAN14, NLE1, FLNC,* and *LIPA*). *GNAQ* and *GNA11* were used as controls (upper panel). mRNA expression (z-scores, lower panel) of the four potential sponges has been compared with two mRNAs highly expressed in patients with a poor clinical outcome (*HTR2B* and *PTP4A3*). The complementarity of the four potential sponges efficiently discriminates the OS of the patients (TCGA cohort). **(C)** Determination of the OS curves by Kaplan–Meier analysis based on the sponge risk model in two cohorts (signature S4). The risk cutoff (low/high) was set to the median of the linear predictor. **(D)** Hypothetical molecular mechanism explaining the loss of tumor

streptomycin (PS) antibiotics (Gibco, Thermo Fisher Scientific). HCT116 cell lines were maintained in McCoy's 5A (Gibco, Thermo Fisher Scientific) supplemented with 10% FBS (EurobioScientific) and 1% penicillin–streptomycin antibiotics (Gibco, Thermo Fisher Scientific). All cell lines have been routinely tested for mycoplasma contamination (Mycoplasma contamination detection kit; rep-pt1; InvivoGen).

### siRNAs and miRNA transfection

Sequences are available in Table S3. All siRNAs and synthetic mimics were transfected at 66 nM using Lipofectamine RNAiMAX (Thermo Fisher Scientific) according to the manufacturer's instructions. For cell density assay: 80,000 cells for 92-1 and HCT116 KO DROSHA, HCT116 WT, and 10,000 for MP41 and 12,000 for 92-1 cells were seeded in 96-well plates, in quadruplicates. For RNA and proteins analysis, 250,000 cells were seeded in six-well plates. Cells were harvested 72 h after transfection (or kinetic). All siRNAs were purchased from IDT DNA. All mimics were purchased from Dharmacon.

### shRNA experiments

Lentiviral particles carrying shRNA vectors targeting human *PYGB* mRNA (shPYGB, TL310025V), and scramble shRNA (shCTR, TR30021V) were purchased from Origen. Lentiviral production was performed as recommended (https://www.epfl.ch/labs/tronolab/) using HEK 293T cells. After infection, cells were maintained in the presence of puromycin 1 μg/ml for selection (Invivogen).

### RNA and miRNA isolation, reverse transcription, and quantitative PCR

RNA was isolated from cell samples using NucleoSpin RNA Plus kit (Macherey-Nagel) and quantified using a NanoDrop 1000 Spectrophotometer (Thermo Fisher Scientific). Reverse transcription was performed with the High Capacity cDNA Reverse Transcription kit (Applied Biosystems). Quantitative PCR was performed on 1 ng cDNA, in 384-well plates using the SYBR Green PCR Master Mix (Applied Biosystems) with the QuantStudio 7 Flex Real-Time PCR System (Applied Biosystems). RNA levels were normalized against human *GAPDH*. Relative amounts of transcripts were determined using the ΔΔ – Ct method and human *GAPDH* transcript level was used as an internal control for each cell line sample.

miRNA was isolated using mirVana miRNA isolation kit (Ambion; Life Technologies). Reverse transcription was performed with the TaqMan microRNA Reverse Transcription Kit (Applied Biosystems) with the Megaplex RT Primers Pool A v2.1 (Applied Biosystems). Quantitative PCR was performed on 1 ng cDNA, in 384-well plates using the TaqMan Gene Expression Master Mix (Applied Biosystems) with the QuantStudio 7

Flex Real-Time PCR System (Applied Biosystems). Relative amounts of transcripts were determined using the Δ Δ – Ct method and human RNU6B was used as an internal control for each cell line sample. The primers were used are described in Table S3.

### Cell-density evaluation

Cell density was measured by methylene blue colorimetric assay (Gilot et al, 2017). Briefly, cells were fixed for 30 min with 70% ethanol. Then, fixed cells were dried and stained 30 min with 1% methylene blue dye in borate buffer. Plates were washed 3 times with fresh tap water and 100 μl of 0.1N HCl per well were added. Plates were analyzed with a spectrophotometer at 620 nm.

### Western blot experiments

Experiments were performed as previously described (Gilot et al, 2017). Membranes (GE HealthCare) were probed with suitable antibodies and signals were detected using the Amersham Imager 680 (Thermo Fisher Scientific). The antibodies are described in Table S3. Uncropped Western blots are available in Source data.

### RNA sequencing

Total RNAs were quantified using a NanoDrop 1000 Spectrophotometer (Thermo Fisher Scientific) and RNA integrity (RIN > 8) was evaluated using RNA nano-chips on the Agilent 2100 Bioanalyzer Instrument (Agilent Technologies). Libraries generation and sequencing experiments have been conducted as previously reported (Corre et al, 2018).

### Biotinylated miRNA pull-down

These experiments were performed on MP41 cell according to the protocol published by Judy Lieberman's laboratory (Tan & Lieberman, 2016) but with minor modifications. 15 millions of MP41 cells were seeded in 3 × 150-mm dishes (5 millions each) and were transfected using Lipofectamine 2000 (Thermo Fisher Scientific) (42 μl per dish) the next day with 100 nM of biotinylated miR-16 or miR-CTR (Dharmacon). Next, they were harvested ~24 h post transfection. Cells from the three dishes were treated separately. Meanwhile, magnetic beads (Streptavidine Dynabeads M-280 DYNAL; Thermo Fisher Scientific) were washed and blocked according to the protocol (Tan & Lieberman, 2016). Cell lysate and washed beads were incubated for 4 h at 4°C on a rotating agitator. All next steps: the precipitation and the purification of coupled RNA was performed according to the protocol. We pooled the three same conditions (from the three transfected plates) in the end of

---

suppressor activity of miR-16 by RNAs (loss of brake effect). miR-16 is considered as a potent tumor suppressor because it regulates the cell cycle by decreasing the expression level of targets such as *CCND3* and *WEE1*. In patient with a poor OS, miR-16 is not able to bind and regulate these RNAs. The sequestration of miR-16 on other mRNAs (defined as potential sponges) is associated with metastasis risk and dismal OS in UM. Instead of promoting RNA decay of miR-16 targets, the non-canonical miR-16 activity promotes expression of potential sponges such as the pro-tumoral PTP4A3 gene (acceleration effect). miR-16 sequestration seems to promote cancer burden by two combined events – "loss of brake and an acceleration." In conclusion, we propose that miR-16 can exert pro- or anti-tumoral activity in function of its base-pairing to mRNAs. For clinicians, our signature S4 accurately predicts clinical outcomes compared with existing classification schemes. Our results expand the current knowledge on molecular mechanisms promoting UM and pave the way to explore new therapeutic candidates targeting miR-16 activity for a cancer without effective treatment at metastatic stage.

the RNA purification. Purified RNA was quantified using a NanoDrop 1000 Spectrophotometer (Thermo Fisher Scientific) followed by the sequencing.

### Biotinylated miRNA pull-down sequencing

RNA sequencing of pull-downed RNA was performed by Novogene according to its RIP sequencing protocol (Illumina PE150/Q30≥80%).

### In silico analyses

The miRNA-binding sites on RNA (MRE) were predicted by webtool TargetScan 7.2 (Agarwal et al, 2015) and RNAhybrid (Rehmsmeier et al, 2004) both available online. Non-canonical MREs have been identified using RNAhybrid and Cistrome SeqPos motif analysis (Liu et al, 2011). From RNAseq data, 903 genes were found to be downregulated (with fold change 1.5). miR-16 peaks falling within those genes were called following the procedure described by Sérandour et al (2012). The resulting bed file containing 504 peaks was used for de novo motif search with the SeqPos tool from Cistrome (Liu et al, 2011), which looked for enriched DNA motifs within these DNA regions. Annotated genes associated with de novo motifs were identified.

To assess the survival prognosis capabilities of the (selected genes or sponges/targets), we performed univariate Cox analyses of the expression data for these genes, with OS as a dependent variable. Patients were divided into two categories according to the median expression of each gene: low expression (below median) and high expression (above median). The Kaplan–Meier method was used to estimate the survival distributions. Log-rank tests were used to test the difference between survival groups. Analyses were carried out with the survival R package.

### Sponge risk model

We used the TCGA-UVM cohort downloaded from the Xena Browser as a training cohort and the GEO dataset GSE22138 as a validation cohort. We trained an optimal multi-gene survival model based on the expression of the sponges in the training cohort by selecting survival-associated genes with the rbsurv R package using 1,000 iterations. Briefly, this package allows a sequential selection of genes based on the Cox proportional hazard model and on maximization of log-likelihood. To increase robustness, this package also selects survival-associated genes by repetition (1,000 times) of a separation between the training and validation sets of samples. Risk scores were determined using classical Cox model risk formulae with a linear combination of the gene expression values weighted by the estimated regression coefficients. The risk cutoff was set to the median of the linear predictor. The Kaplan–Meier method was used to estimate the survival distributions. Log-rank tests were used to test the difference between survival groups. Analyses were carried out with the survival and survivalROC R packages.

### Statistics and reproducibility

Data are presented as mean ± SD unless otherwise specified, and differences were considered significant at a $P$-value of less than 0.05. Comparisons were performed using bilateral Student test (with non-equivalent variances). All statistical analyses were performed using Prism 8 software (GraphPad) or Microsoft Excel software.

OS was estimated using the Kaplan–Meier method. Univariate analysis using the Cox regression model or log-rank test, as specified, was performed to estimate hazard ratios and 95% confidence intervals. All experiments were performed three or more times independently under similar conditions, unless otherwise specified in the figure legends (raw data available in Table S4).

## Data Availability

Further information and requests for resources and reagents should be directed to and will be fulfilled by David Gilot (david.gilot@univ-rennes1.fr). All unique/stable reagents generated in this study are available from the Lead Contact with a completed Materials Transfer Agreement. All other data supporting the findings of this study are available from the corresponding author on reasonable request. The human melanoma data set (UM, IlluminaHiSeq) was derived from the TCGA Research Network: http://cancergenome.nih.gov. The data set derived from this resource that supports the findings of this study is available at https://genome-cancer.ucsc.edu.

mRNAseq and RIPseq data that support the findings of this study have been deposited in the Gene Expression Omnibus under accession code GSE180399 (https://www.ncbi.nlm.nih.gov/geo/query/acc.cgi?acc=GSE180399) and ArrayExpress under accession code E-MTAB-10940 (https://www.ebi.ac.uk/arrayexpress/experiments/E-MTAB-10940/).

## Supplementary Information

## Acknowledgements

The authors thank the Gene Expression and Oncogenesis team from the Centre National de la Recherche Scientifique (CNRS) UMR6290, Dr Pascal Loyer from NuMeCan (INSERM U1241), BIOSIT core facilities of Rennes 1 University (SFR UMS CNRS 3480 – INSERM 018, especially P Gripon for the BSL3), the UCA GenomiX platform of Institut de Pharmacologie Moléculaire et Cellulaire, and the Centre de Ressources Biologiques humaines Santé (especially C Pangault) for their help. The authors thank Dr FA Karreth, Dr M Migault, Pr Marc-Henri Stern, Sylvain Martineau, Pr Stéphan Vagner, Pr Judy Lieberman, Dr Karla Meza Sosa, and Dr Shen Mynn Tan for helpful discussion. Support was provided by a "Ligue Nationale Contre le Cancer" (LNCC) fellowship and French Ministry of Research ("Ministère français de l'Enseignement supérieur, de la Recherche et de l'Innovation") fellowship (AM Quéméner). The authors are grateful to Narry Kim for providing the HCT116 KO DROSHA and HCT116 WT (Korean Collection for Type Cultures [KCTC]) and to D Decaudin for the MP41 cell line. This study received financial support from the Ligue Nationale Contre le Cancer (LNCC) Départements du Grand-Ouest; Fondation ARC pour la Recherche; AVIESAN Plan Cancer, Région Bretagne; University of Rennes 1; CNRS; and Ministère de la Recherche et de

l'Enseignement Supérieur and Rennes Métropole. The authors thank La ligue Contre le Cancer.

## Author Contributions

AM Quéméner: conceptualization, software, formal analysis, validation, investigation, visualization, methodology, and writing—original draft, review, and editing.
L Bachelot: formal analysis, validation, investigation, visualization, methodology, and writing—original draft, review, and editing.
M Aubry: resources, data curation, software, formal analysis, investigation, visualization, methodology, and writing—review, and editing.
S Avner: resources, software, formal analysis, visualization, methodology, and writing—review and editing.
D Leclerc: formal analysis, investigation, and writing—review and editing.
G Salbert: resources, software, visualization, and writing—review and editing.
F Cabillic: investigation and writing—review and editing.
D Decaudin: resources and writing—review and editing.
B Mari: resources, formal analysis, visualization, and writing—review and editing.
F Mouriaux: writing—original draft, review, and editing.
M-D Galibert: funding acquisition, project administration, and writing—review and editing.
D Gilot: conceptualization, software, supervision, funding acquisition, investigation, visualization, methodology, project administration, and writing—original draft, review, and editing.

## Conflict of Interest Statement

The authors declare that they have no conflict of interest.

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
