## [Reviewer comments · Life Science Alliance]

Life Science Alliance

Non-canonical miRNA-RNA base-pairing impedes tumor suppressor activity of miR-16

Anais Quemener, Laura Bachelot, Marc Aubry, Stéphane Avner, Delphine Leclerc, Gilles Salbert, Florian Cabillic, Didier Decaudin, Bernard Mari, Frederic Mouriaux, Marie-Dominique GALIBERT, and David Gilot

DOI: <https://doi.org/10.26508/lsa.202201643>

Corresponding author(s): David Gilot, University of Rennes 1

Review Timeline:

Submission Date:	2022-08-02
Editorial Decision:	2022-09-12
Revision Received:	2022-09-13
Accepted:	2022-09-15

Transaction Report:

Please note that the manuscript was reviewed at Review Commons and these reports were taken into account in the decision-making process at Life Science Alliance.

1st Authors' Response to Reviewers

First, we would like to sincerely thank the managing Editor and the referees for acknowledging the impact of the study and for their comments as well as constructive suggestions, which help us to improve our study.

We acknowledge that the letter format of our manuscript was not appropriated. So, as recommended by all reviewers, the revised version is a regular format (full-length MS).

We have answered to all of their queries.

Reviewer #1

(Evidence, reproducibility and clarity (Required)):

Summary: The article presents interesting data regarding the role of miR-16 in the development and progression of uveal melanoma. The authors propose the analysis of miR-16 activity as a marker for uveal melanoma progression instead of miR-16 expression. To analyze the activity of miR-16 they propose a risk model developed around a 4 genes signature that was shown to be able to stratify uveal melanoma patients into low and high-risk groups. Unfortunately, the current version of the manuscript is hard to understand and to carefully evaluate due to a lack of structure and clear presentation of the experimental design, methods, results, and discussions.

We sincerely thank the reviewer for he/his careful review and constructive comments

Major Comments:

- The manuscript needs to be restructured into clear sections including a detailed introduction, materials and methods, results, discussions, and conclusion. The present format is hard to follow with information and results being spread across multiple documents and parts of the manuscript.

We agree with this comment. We modified the manuscript format: from a letter (2 figures) into a classical manuscript (full length) to improve the understanding.

- The authors mention that the level of miR-16 reached after transfection is higher than that in the physiological level, but there is no data to support this. I recommend adding a quantification of miR-16 levels achieved after transfection as part of the S2.

As recommended, we quantified the miR-16 expression level after ectopic expression of this miRNA. A RT-qPCR quantification has been done and results are available on Fig 1H.

- The way of using references is confusing as it is not clear what was done in the results section and what work is cited from the literature. The results section should focus strictly on presenting the results achieved by the experiments while delineating clearly the work that was done in other articles.

As suggested, we payed attention to distinguish our own data and previously published data in the revised manuscript.

- The discussion section needs to be extended to better present the role of specific investigated genes and proteins like PYGB and PTP4A3 in the development and progression of uveal melanoma.

The discussion section has been upgraded with the role of *PTP4A3* in UM.

- The experiments analyzing the sequestration of miR-16 at non-canonical sites are performed using the cell line HCT116 WT and DROSHA KO. HCT116 is a human colon adenocarcinoma cell line, a tumor with a completely different histology. These experiments should be performed also on a human uveal melanoma cell line in order to ensure consistency of the results.

In this experiment, our goal was twice.

Firstly, we wanted to evaluate the miR-16 ability to upregulate RNA/protein expression *via* non-canonical binding sites. This type of RNA upregulation might also be the result of a competition between two miRNAs that are able to bind the same sequence (miR-16 and another one). So, we investigated if miR-16 prevents the binding of another miRNA (which promotes the RNA turn over). To evaluate such scenario, we selected an already published DROSHA KO cell line, **in which almost all miRNAs were lost due to the DROSHA KO** (Kim et al., 2016), including miR-16. Using this cell line and a miR-16 rescue (with synthetic miR-16 transfection), we showed that miR-16 binds directly to non-canonical MRE-16 to promote translation. We confirmed this conclusion using a mutated binding site for miR-16 (Fig 4E-F).

Secondly, we wanted to validate this miR-16 ability to promote gene expression in another cancer model. In HCT116 (adenocarcinoma) cells, we confirmed in figure 4B that miR-16 itself promotes mRNA upregulation (as observed in 3 UM cell lines, Figure 4A).

During these experiments, we also generated a DROSHA KO cell line (MP41) in order to reproduce these experiments in an UM model (see the figure below). We selected the same sgRNAs than Darnell's team (Luna et al., 2015) (Annexe 1A) and we successfully established a new cell line; DROSHA KO MP41 (validated by western-blot and genomic analyses (showing a short deletion in *Drosha* gene) (Annexe 1B-C)). These results are strictly the same than previously published (Luna et al., 2015).

Annexe 1:

A

B

C

D

E

F

However, the miR-16 quantification was disappointed in MP41 DROSHA KO cells because miR-16 expression level was still detectable (Annexe 1D) in contrast to three DROSHA KO cells (HCT116, Huh-

7.5 and HEK293T DROSHA KO cells) established by (Dai et al., 2020; Kim et al., 2016; Luna et al., 2015).
So, we couldn't use this line despite our efforts.

We are currently investigating the reasons. We recently analyzed the expression levels of 89 miRNA in our DROSHA KO MP41 versus the DROSHA KO Huh-7.5 cells (Annexe 1E-F). We confirmed that 75% of miRNAs disappeared in DROSHA KO Huh-7.5 cells and 11% are stable despite the loss of DROSHA. In MP41 cells, the loss of DROSHA had not the same effect (only ~26% of these miRNAs disappeared). These results suggested that the miRNA processing might differ in function of the cell type or cancer cells.

Minor Comments: • The manuscript is lacking consistency regarding the usage of abbreviations. These should be defined the first time when used in the text.

We apologize. It has been improved.

• The figure S2 needs to be adjusted. It is hard to understand in S2B how the statistical analysis was performed. I recommend representing each line with the two conditions side by side to increase clarity.

We agree. The graphs were updated as suggested.

• When presenting the miR-16 interactome the data are spread in three different sources Figures S2, 1D-E, and Table S1 which makes it difficult to follow the images. I recommend presenting these data in the same figure.

We agree. Please see the new figure.

• The manuscript requires English grammar and style editing. There are several words misspelled and phrases with a complicated syntax that makes it difficult to understand.

We apologize. It will be improved using an editing service.

• The method of presenting the data in Figures 1 F and H needs to be reorganized. The current version makes it very hard to understand how the gene expression changed after miR-16 exposure.

We agree and we improved it. We tried, as suggested, another graphical view (heatmap) but we believe that the data are less understandable.

Reviewer #1 (Significance (Required)):

The article presents important results regarding the role of miR-16 in uveal melanoma by an innovative approach analyzing the activity of miR-16 instead of its expression. The authors focus on the relationship between miR-16 sponges and targets and through a set of elegantly designed experiments they identify a set of 4 genes whose expression can be used as a risk predictor model in uveal melanoma.

The audience of this article can be represented by both clinicians and researchers that could take advantage of the results presented. Also, the approach of the article can open new directions for other cancers and miRNAs.

We thank this reviewer for her/his positive comments on our work and for the suggestions made to improve its relevance

(Evidence, reproducibility and clarity (Required)):

This is an extremely short report that identified a potentially new mechanism as to how miR-16 may be involved in uveal melanoma (UM). In this report the authors used previous data, that identified that miR-16 is involved in UM, to gain a more comprehensive understanding of the mechanism involved. miR-16 was selected as a candidate due to its location in chromosome 3, which UM patients of the have chromosome 3 monosomy. The group evaluated the transcriptome following miR-16 overexpression in a single cell line, and as others often find with miRNAs, there was a cohort of up- and down-regulated RNAs. Figure 1 is a summary of the workflow, indicating the number of up- and down-regulated RNAs with validations performed. In figure 2 the authors identified 2 cohorts based on the previous expression data, one of which defines a more at-risk group. While the data presented in Figures 1 and 2 is interesting overall, the conclusions are mostly drawn from a supplemental figure. The data in this manuscript needs to be reevaluated and the most critical included in the main figures. If done so, and rewritten appropriately this study could have a bigger impact. As is, this study would be of low to moderate interest. There are also some instances, early on, where conclusions are overstated. Overall the text is lacking in description to conduct a thorough review, the authors fail to provide an introduction to put the study in the context of the field, and they provide a one-sentence discussion. Some of these issues are defined below, but due to the lack of description of the studies, and overstatements made, this is not a thorough review.

We sincerely thank the reviewer for her/his careful review and constructive comments. As explained in the introduction of this letter, the manuscript format has been changed; letter into regular full-length MS.

Major issues:

The main text and Figure 1 inappropriately referred to upregulated RNAs as miR-16 sponges. At this point in the manuscript, these are nothing more than upregulated RNAs.

We agree. It has been modified in the revised version. These genes are now called "UP-regulated genes".

The text is often vague and lacks discretion that is essential for the Reviewer to understand the study. Some sentences are fragments and are not clear. Some (but not all) examples of difficulties encountered while attempting the review are indicated below as well.

As recommended, the manuscript format is now a regular full-length MS to overcome the lack of details.

Figure 1E legend. "in function of the experimental workflow detailed". "In function" does not make sense in this sentence. It is not clear what the authors are referring to. Similarly, "In function of their expression..." In function aging is inappropriate and the sentence is not clear. Also "MRE for miRNA response element". Perhaps the authors mean "MRE = miRNA response element." Also, the text "10arbouring" is included in this legend. In brief, Figure 1E is extremely difficult to evaluate with the poor text in the legend.

We apologize. In the revised version, the style and the grammar have been improved using an editing service.

Multiple figures have odd wording to indicate biological replicates. This needs to be clarified in better, complete sentences.

We apologize. The submitted files contained all the raw data (Table S4). This file contains all the replicates and the number of experiments per graph. To help the readers, we improved the wording about the biological replicates in the revised version (legends).

For Figure S2, Fold induction is indicated as a %. This is not appropriate. It is either a fold change or a change in %, but not both.

We agree. We revised this mistake in the revised version of the manuscript.

How is the experiment done in Figure S3A a "kinetic" experiment? There is no kinetic analysis here.

We thank the reviewer for this comment. This point has been fixed in the revised version of the manuscript. In fact, the kinetic was shown in Fig. 1D. For Fig. S3A, we chose the 24h and 48h data (indicated as mRNAseq on Fig.3A). The idea was to identify the main canonical targets of miR-16.

Details of the critical biotinylating study are completely lacking. And since this is the most critical experiment that gets to the main point of the study, this should be well defined and part of the main figures in the manuscript. It should not be in the supplemental information. All of Figure S2 should be in the main part of the manuscript.

We thank the reviewer for this suggestion. This experiment is now with the main figures (Fig 2). For the RNA-IP protocol, we strictly followed the protocol provided by Prof. Judy Lieberman published in 2016 (Tan and Lieberman, 2016). The protocol was available in Supplementary Information & Experimental Procedures file (page 4, L92-L104). Moreover, we thank these authors for helpful advices (see section acknowledgments). In the revised version, we explained in more details this protocol including the biotinylated miRNA sequences.

All axis in Figure S2B are not labeled appropriately. Enrichment is used, however not all RNAs are enriched. Perhaps fold change would be a better and more accurate name. Those on the left side (in blue) are depleted, not enriched.

On this plot, we showed for each gene : its RNA expression after miR-16 transfection (x-axis; log FC) and its binding to biotinylated miR-16 (y-axis, now labeled reads count in biotinylated miR-16 condition). In fact, the Enrichment term was removed and replaced by "reads count" in biotinylated miR-16 condition.

mRNAs highlighted in the blue box are highly associated with miR-16 and their expression levels decreased in response to this interaction. We improved these sentences in the revised version.

For Figure S3C, the authors should change "blue ones" to "solid blue bars indicate". Same for S3E

We agree. We corrected this mistake in the revised version of the manuscript.

For Figure S3D, "logo" is inappropriate and not the correct term. This is a consensus sequence, not a logo.

We revised this term in the revised version of the manuscript. The term Logo was initially used in the *editio princeps* (Liu T, et al. Genome Biol. 2011. PMID: 21859476).

How did the authors make the conclusion that the upregulated RNAs are targets of miR-16 if they do not have a canonical miR-16 binding sites? They could easily be indirect RNAs that are elevated post miR-16 exposure. The authors do not validate that the cohort of RNAs upregulated are indeed miR-16 targets. Thus, the overstatement of "sponge" RNAs (Figure 1H) or even "target" RNAs (Figure 1F) without appropriate validation is overstated. Simply doing the biotinylating study is still not enough to conclude direct interactions of these RNAs with miR-16. These can be false positives, that are not well controlled for due to poor selection of a control RNA.

We thank the reviewer for the comment. We agree that the sponge term is overstated here. We modified it in the revised version. We also added a new paragraph in the discussion to explain the limits of our study (especially this notion). To be honest, few authors validate by luciferase assays their all potential miR-targets (as done for figure 6E in (Lal et al., 2011)), authors usually assume that a decrease of mRNA in response to miRNA might be a target of this miRNA if the mRNA contains a consensus miRNA binding site. We agree, a correlation is not a proof of a direct effect. So, we used the following terms in the revised MS: 'potential targets', potential sponge RNAs', down-regulated RNAs and up-regulated RNAs to avoid/limit overstatement.

Our workflow was designed to identify "potential mRNA-miR-16 interaction" (miR-16 pull down) without *a priori*. We also mentioned in the result section that non-specific binding might occurred in our RNA-IP experiment as always seen in whole genome experiments. To reduce the risk, we used biotinylated

miRNA control and we performed validation experiments based on different approaches (miRNA, shRNA, luciferase assay, KO cells). Importantly, these RNAs (candidate sponges) are able to predict the overall survival of patients with UM. To the best of our knowledge, this is the first time that a predictive signature is composed of genes belonging to the same mechanism (miR-16) for UM. So, it is tempting to conclude that UM is both a GPCR & a miR-16 disease.

To conclude, we concede that our workflow is probably not perfect but the results are useful for clinicians and we made our best.

Figure 1G is not large enough to see, nor is the inclusion of it clear in the text of the manuscript.

We thank the reviewer for the comment. We improved this item in the revised version.

Figure 1H, referring to upregulated RNAs, post miR-16 expression as sponges is inappropriate unless they have all been validated as miR-16 sponges. These could merely be RNAs that are indirectly upregulated following miR-16 transfection, and their upregulating following miR-16 overexpression has been validated. However, their miR-16 sponging activity has not been validated. Similarly for Figure 2A.

We agree. At this step, the term sponge is inappropriate. We changed it by UP regulated genes.

Figure 2 is poorly defined. This needs clarification and the font should be increased. There are also "..." in the figure legend which is inappropriate. Many things are not defined such as "CN" which the reviewer is assuming means copy number. Also the colors and description for "Yes/Dead/Male" are not clear. What are these? How are they relevant?

We thank the reviewer for these comments. We improved the size in the revised version.

Since we analysed many parameters on the same heatmap, a colour has been used for different parameters. Yes/Dead/Male (black colour), parameters associated to Metastasis, Vital status and gender. Black colour means a value of 1 or Yes. By default, white colour means a value of 0 or No. This description (black/white) avoids the use of many colours that colour-blind people are unable to see. I am colour-blind. (see TIPS AND TOOLS published in Nature 04 October 2021, <https://www.nature.com/articles/d41586-021-02696-z>).

These parameters are important for the conclusion.

For Figure 2B legend, what is meant by miR-16 "in function"?

We corrected this mistake. In the revised version of the manuscript, you can read : Boxplot comparing the amount of miR-16 in patients who developed metastases or not (TCGA cohort).

The authors should show the level of upregulation of miR-16 following transfection for all experiments where miR-16 is transfected.

As recommended, miR-16 quantifications by RT-qPCR have been added to the main figures.

For all figures where qRT-PCR was conducted, what are RNAs normalized to. This should be indicated on the axis and/or in the figure legend. While in the methods section, this should also be present in the main body of text (ie. figures).

We thank the reviewer for this comment. This point has been fixed in the revised version of the manuscript. We always use GAPDH for mRNAs and U6 for miRNAs.

For Sup Figure 2A the authors indicate that RNA levels were compared to 501Mel. They should show the 501mel levels in the same graph. They also state that the absolutely copy number was determined from Northern Blot. As the authors likely know, quantification using qRT-PCR is much more quantitative than Northern. They should conduct qRT-PCR for the main cell line they are comparing to. The Northern is also not shown and the reference provided for it is for a Nature Review article, not for a study that shows a Northern blot.

We thank the reviewer for this comment. In our previous study published in 2017 in Nature Cell Biology (Gilot, Migault et al. 2017), we performed Northern blot experiments to quantify the miR-16 expression levels according to reviewer comments. In page 3, line 96 we cited reference 11. In fact, it was the reference 8. We will modify it in the revision version. We apologize.

As suggested, we included the qRT-PCR for all cell lines.

It is not clear what the control RNA was for all the studies. Specifically for the biotinylated studies, the authors should use another miRNA, not a non-specific control. Because a control miRNA will also binds AGO and other miRNA-associated factors, non-specific binding due to these factors could be better controlled for. The non-specific RNA will not account for these factors.

We thank the reviewer for this comment. The miRNA control has been provided by Dharmacon (Reference NC2, miRIDIAN microRNA Mimic Negative Control #2). The RNA sequence is UUGUACUACACAAAAGUACUG (MIMAT0000295), cel-miR-239b, mature sequence. We used the same reagents and miRNA control than published by Tan S & Lieberman J, 2016).

In the literature, we did not find a gold-standard control for such experiments. In function of the authors, they used a control miRNA, or another miRNA, a scramble sequence or a miRNA from another species. We also believe that non-specific RNAs binding is mainly due to the beads.

Sup Fig 4 is missing details. What orientation is the consensus sequence shown in relative to the miRNA (5'-3' or 3'-5')? Other details are missing as well, this is just one example of many issues.

We apologize. It has been improved in the revised version.

For Sup Fig 5A the CT values should be included. That gives a better direct comparison than a graph of something that is indicated as not determined. You cannot graph something that is not determined.

We thank the reviewer for the comment. Since we confirmed data already published, we removed this item on the revised version.

For Sup Fig 5C the font cannot be read it is too small.

We improved in the revised version

For Sup Fig 5D, again, how much miR-16 is present when overexpressed. Would this amount be physiologically achievable?

The miR-16 quantification has been included into the revised version. In fact, we know as others researchers that miRNA or plasmid quantification (post transfection) is imperfect because a large amount of miRNA or plasmid stays in 'cellular compartments' after transfection. These "isolated small RNA or plasmids" are detected by qRT-PCT after a cell lysis but they are not able to act into the cell. We published these observations in 2002 (Gilot et al., 2002) using plasmids coupled to fluorochromes. Similar results have been yet published for siRNA and miRNA.

Based on the qRT-PCR, the ectopic expression of miR-16 is probably not physiological. On the submitted MS, page 3 lines 94-96, we clearly mentioned this limit. In the other hand, we performed experiments without miRNA transfection. The *PYGB* depletion demonstrated that sequestered (endogenous) miR-16 on *PYGB* can be redirected toward it targets (previously Fig. S5 E-F). In this case, only endogenous miR-16 is able to regulate these targeted RNAs.

The title is poor and not descriptive enough for the study. It reads more like the title for a review article.

We agree. The title has been changed. Non-canonical miRNA-RNA base-pairing impedes tumor suppressor activity of miR-16.

Methods for siRNAs indicated kinetic as well. Not clear what kinetic data were acquired during this study.

We thank the reviewer for the comment. It has been corrected in the revised version.

Reviewer #2 (Significance (Required)):

****Referees cross-commenting****

I am in full agreement with the additional comments made by Reviewer #1. I however disagree with Reviewer #3 that the study was well conducted and "elegant". Based on multiple issues (many cooperated) between R#1 and R#2 I do not feel that this study is acceptable and will take an extraordinary amount of time to be acceptable for publication.

We thank this reviewer for her/his positive comments on our work and for the suggestions made to improve its relevance

(Evidence, reproducibility and clarity (Required)):

Uveal melanoma is the most common primary intraocular malignancy in adults. About 50% of patients develop metastases, being the liver the most common place for them. Despite over 50 years of study, little progress has been done in efficacious treatments. In this report, the authors aimed at a better understanding of the mechanistic drivers for cancer aggressiveness and poor overall survival at the metastatic stage. To accomplish this, the authors performed a series of elegant genomics and transcriptomics analyses and identified molecular aberrations in miR16, which has been previously associated with other malignancies. The authors demonstrated that high level of miR16 sponges inversely correlate with poor overall survival. As a reference and validation, they are using the TCGA data analysis. Lastly, the authors generated a signature for survival prediction based on 4 genes, which was confirmed using an independent study. The methodology was very elegant. The appropriate analyses were done.

Reviewer #3 (Significance (Required)):

This manuscript provides incremental knowledge to the field. In the last 5 years there are many manuscripts addressing different transcription factors or miRNA molecules and their role in different cancers. Uveal melanoma is an orphan disease with high unmet need. Prognostication is highly valuable, however; it is the treatment where we need the most attention.

The authors did elegant studies to demonstrate the relevance of miR16. This is not part of the standard of care, but the prognostic tool of the selected 4 genes, could be very helpful. I wish they could have included a sentence on the impact in the field.

We sincerely thank the reviewer for he/his careful review and constructive comments. We added a paragraph in the discussion section about this 4-genes signature and the others, especially the recent works done by the Bertolotto's team (Strub T. et al., 2021 and Pandiani C. et al., 2021).

The response to the following questions can make the manuscript more robust: Description of the TCGA - how many of the primary tumors had clinically detectable metastases? This is important as you are describing a potential companion diagnostic testing to predict OS.

We thank the reviewer for the comment. This information was available in Robertson's MS published in Cancer Cell, 2018 (TCGA, supplementary Tables). On the Fig 5A, the number of patients who developed metastases was indicated (metastasis: black colour = Yes). So, 26 mets. Using our sponge signature, we predicted that 23 patients are associated to bad clinical outcome (metastases).

We need additional information on the UM cells, which can be found in literature. It is necessary for the audience to understand which of these cell lines come from patients that die from metastases, which ones had additional malignancies. There are UM cell lines - commercially available ? for which the primary and metastatic line were developed from the same patient. Those are very helpful, especially if one of the objectives is demonstrating poor OS due to metastatic disease. That was not clear from the manuscript.

We thank the reviewer for the comment. The characterization of these cell lines has been added in the results section. These cell lines are commercially available.

Levels of miR16 - What were the copy numbers in healthy patients? Do we need them relative to 501 Mel? or should we compare to a system that is not dysregulated?

We agree, it is an important query. We showed no miR-16 difference between patients with a poor or good OS in the TCGA cohort. This result confirmed that the expression level of miR-16 is mainly unchanged during the tumor progression.

In the revised version of the manuscript, we quantified the expression level of miR-16 in melanocytes in primary cultures versus 501Mel cells. The expression level of miR-16 is less important in normal melanocytes when compared to melanoma cell lines. Data have been added in Fig 1E.

Why were the DROSHA KO be lung cancer cells? HCT116? Why choosing this cell line?

In this experiment, our goal was twice.

Firstly, we wanted to evaluate the miR-16 ability to upregulate RNA/protein expression *via* non-canonical binding sites. This type of RNA upregulation might also be the result of a competition between two miRNAs that are able to bind the same sequence (miR-16 and another one). So, we investigated if miR-16 prevents the binding of another miRNA (which promotes the RNA turn over). To evaluate such scenario, we selected an already published DROSHA KO cell line, **in which almost all miRNAs were lost due to the DROSHA KO** (Kim et al., 2016), including miR-16. Using this cell line and a miR-16 rescue (with synthetic miR-16 transfection), we showed that miR-16 binds directly to non-canonical MRE-16 to promote translation. We confirmed this conclusion using a mutated binding site for miR-16 (Fig 4E-F).

Secondly, we wanted to validate this miR-16 ability to promote gene expression in another cancer model. In HCT116 (adenocarcinoma) cells, we confirmed in figure 4B that miR-16 itself promotes mRNA upregulation (as observed in 3 UM cell lines, Figure 4A).

During these experiments, we also generated a DROSHA KO cell line (MP41) in order to reproduce these experiments in an UM model (see the figure below). We selected the same sgRNAs than Darnell's team (Luna et al., 2015) (Annexe 1A) and we successfully established a new cell line; DROSHA KO MP41 (validated by western-blot and genomic analyses (showing a short deletion in *Drosha* gene) (Annexe 1B-C)). These results are strictly the same than previously published (Luna et al., 2015).

Annexe 1:

However, the miR-16 quantification was disappointed in MP41 DROSHA KO cells because miR-16 expression level was still detectable (Annexe 1D) in contrast to three DROSHA KO cells (HCT116, Huh-7.5 and HEK293T DROSHA KO cells) established by (Dai et al., 2020; Kim et al., 2016; Luna et al., 2015). **So, we couldn't use this line despite our efforts.**

We are currently investigating the reasons. We recently analyzed the expression levels of 89 miRNA in our DROSHA KO MP41 versus the DROSHA KO Huh-7.5 cells (Annexe 1E-F). We confirmed that 75% of miRNAs disappeared in DROSHA KO Huh-7.5 cells and 11% are stable despite the loss of DROSHA. In MP41 cells, the loss of DROSHA had not the same effect (only ~26% of these miRNAs disappeared). These results suggested that the miRNA processing might differ in function of the cell type or cancer cells.

Overall, I consider this a good manuscript and with some tweaking it can be better.

We thank this reviewer for her/his positive comments on our work and for the suggestions made to improve its relevance

****Referees cross-commenting****

Based on the different reviews and the added note, we all agree the manuscript needs work and is not ready yet to be accepted. We all agreed that needs to be re-structured as we all mentioned about key missing pieces in the writing. It will be of help to the authors to go back to the guidelines to know the max words and extend their manuscript.

The timing needed is might not be too relevant, as we all agreed it needs work. Thank you to my reviewers/colleagues as they pointed out things very comprehensively. Agreed with their comments.

September 12, 2022

RE: Life Science Alliance Manuscript #LSA-2022-01643

Dr. David Gilot
CNRS UMR6290 - University of Rennes1
Rennes 35043
France

Dear Dr. Gilot,

Thank you for submitting your revised manuscript entitled "Non-canonical miRNA-RNA base-pairing impedes tumor suppressor activity of miR-16". We would be happy to publish your paper in Life Science Alliance pending final revisions necessary to meet our formatting guidelines.

- please add a running title, alternate abstract, and a category to our system
- please add the Twitter handle of your host institute/organization as well as your own or/and one of the authors in our system
- please use the [10 author names, et al.] format in your references (i.e. limit the author names to the first 10)
- please double-check the panels in your Figure 6 legend and label the panel E as panel D, and add a figure callout for Figure 6D to your main manuscript text
- please rename your EV figures as supplementary figures and adjust the figure callouts in the main manuscript text accordingly. Figure EV4 can be uploaded and labeled as Source Data
- please title your figures with their figure number in our system
- please use a single Data Availability Statement. Some of the information in the AVAILABILITY of DATA and MATERIALS section is redundant, and this separate section should be removed.
- The ETHICAL APPROVAL AND CONSENT TO PARTICIPATE and CONSENT for PUBLICATION sections should be removed

A. FINAL FILES:

B. MANUSCRIPT ORGANIZATION AND FORMATTING:

Sincerely,

Reviewer #1 (Comments to the Authors (Required)):

The authors revised the manuscript according to the comments.

This is a new mechanism of tumorigenesis, and the results are of great interest for the readers of the journal.

September 15, 2022

RE: Life Science Alliance Manuscript #LSA-2022-01643R

Dr. David Gilot
University of Rennes 1
INSERM U1242 - CNRS UMR 6290
Rue de la Bataille Flandres Dunkerque
CS 44229
Rennes 35043
France

Dear Dr. Gilot,

Thank you for submitting your Research Article entitled "Non-canonical miRNA-RNA base-pairing impedes tumor suppressor activity of miR-16". It is a pleasure to let you know that your manuscript is now accepted for publication in Life Science Alliance. Congratulations on this interesting work.

DISTRIBUTION OF MATERIALS:

Again, congratulations on a very nice paper. I hope you found the review process to be constructive and are pleased with how the manuscript was handled editorially. We look forward to future exciting submissions from your lab.

Sincerely,
